# Behavioral and physiological correlates of kinetically tracking a chaotic target

**Atsushi Takagi** [1,2,3]*, **Ryoga Furuta**[4], **Supat Saetia**[2], **Natsue Yoshimura**[2,3], **Yasuharu Koike**[2], **Ludovico Minati**[2,5]

**1** NTT Communication Science Laboratories, Atsugi, Japan, **2** Institute of Innovative Research, Tokyo Institute of Technology, Yokohama, Japan, **3** Precursory Research for Embryonic Science and Technology (PRESTO), Japan Science and Technology Agency (JST), Kawaguchi, Japan, **4** Department of Information and Communications Engineering, School of Engineering, Tokyo Institute of Technology, Yokohama, Japan, **5** CIMeC, Center for Mind/Brain Sciences, University of Trento, Trento, Italy

* atsushi.takagi.yx@hco.ntt.co.jp

**Data Availability Statement:** Data is on a public repository figshare (10.6084/m9.figshare.12253649).

**Funding:** A.T. and L.M. received funding from the World Research Hub Initiative (WRHI), Institute of

## Abstract

Humans can innately track a moving target by anticipating its future position from a brief history of observations. While ballistic trajectories can be readily extrapolated, many natural and artificial systems are governed by more general nonlinear dynamics and, therefore, can produce highly irregular motion. Yet, relatively little is known regarding the behavioral and physiological underpinnings of prediction and tracking in the presence of chaos. Here, we investigated in lab settings whether participants could manually follow the orbit of a paradigmatic chaotic system, the Rössler equations, on the (x,y) plane under different settings of a control parameter, which determined the prominence of transients in the target position. Tracking accuracy was negatively related to the level of unpredictability and folding. Nevertheless, while participants initially reacted to the transients, they gradually learned to anticipate it. This was accompanied by a decrease in muscular co-contraction, alongside enhanced activity in the theta and beta EEG bands for the highest levels of chaoticity. Furthermore, greater phase synchronization of breathing was observed. Taken together, these findings point to the possible ability of the nervous system to implicitly learn topological regularities even in the context of highly irregular motion, reflecting in multiple observables at the physiological level.

## Introduction

A remarkable property of nonlinear dynamical systems is their ability to generate highly complex trajectories in spite of structural simplicity. From the canonical three-body problem through toy models such as the double pendulum and the dripping water faucet, chaotic motions permeate nature even in the most unsuspecting circumstances. Far from representing randomness, they combine *de facto* unpredictability with well-evident and elegant topological regularities, challenging at heart the ancestral notions of determinism [1]. The profound influence of chaos theory on contemporary science can be gauged by the fact that universal features of nonlinear dynamics are cohesively observed across systems as diverse as meteorological

Innovative Research (IIR), Tokyo Institute of Technology, Tokyo, Japan. A.T. was partially supported by the JST PRESTO (Precursory Research for Embryonic Science and Technology) grant JPMJPR18J5. A.T, N.Y. and Y.K. were partially supported by JST Mirai grant JPMJMI18C8. N.Y. was partially supported by the JST PRESTO grant JPMJPR17JA. The funders had no role in study design, data collection and analysis, decision to publish, or preparation of the manuscript.

**Competing interests:** The authors have declared that no competing interests exist.

phenomena, geophysical events, electronic circuits, and neural dynamics from the single-cell level up to entire brains [2–4]. Unsurprisingly, chaotic dynamics also spontaneously emerge in physiological rhythms such as heart rate variability and gait generation [5–7]. The central nervous system could, therefore, have evolved an innate ability to predict, generate and possibly control chaos. Here, the question is approached from the physical perspective of a motor control task.

To date, the relationship between chaotic dynamics and human behavior has only been considered in a limited number of studies. Some have examined the volitional control of a process governed by chaotic motion, whereas others have illuminated the ability to judge randomness versus chaoticity and predict, or synchronize to, the temporal evolution of discrete steps such as the logistic map, or continuous flows [8–12]. Collectively, these existing works point to an ability of gradually attaining higher-than-chance performance during exposure to a chaotic trajectory. However, to the authors' knowledge, the effect of the "level of the chaoticity", practically reflecting in irregularity and unpredictability, has not yet been explicitly addressed in terms of the behavioral and physiological correlates of attempting to physically track the motion of a target. This setting appears particularly pertinent, given the ecological relevance of successfully chasing, grasping or avoiding an erratically-moving object.

In this work, we consider the paradigmatic case of a low-dimensional chaotic system, namely the Rössler equations, which generate a predominantly circular orbit on the $(x, y)$ plane while twisting and spiking along the $z$ dimension. For increasing settings of the control parameter $a$, it can give rise to more prominent irregularity and folding reminiscent of a Möbius strip, which manifest as transients in a particular region of the phase space. We manipulated this control parameter over the range spanning a trivial circular closed orbit through fully-developed chaos, while monitoring multiple kinematic parameters of arm movement alongside electromyographic (EMG), electroencephalographic (EEG) and peripheral physiological activity.

We hypothesized that the participants would be able to track the irregular folding of the chaotic orbits with an accuracy beyond chance level, possibly by implicitly learning a predictive model of its dynamics [13, 14], and that their tracking accuracy would depend on the chaoticity level. We also anticipated that muscular co-contraction would correspondingly decrease with practice [15, 16] as the participants adapted their behavior to track the target with less effort. Neural activity, namely the EEG rhythms, could also respond similarly, as elevated attention and readiness are prerequisites for tracking and attempting to predict a complex motion [17]. We furthermore anticipated that this task could bring about entrainment effects at the level of respiration and heart rate, as previously observed in other contexts [18–20].

## Materials and methods

### Experimental apparatus

The procedures in this study were approved by the Institutional Review Board of the Tokyo Institute of Technology (no. 2017142, 30 March 2018, P.I. N.Y.). All procedures performed in studies involving human participants were in accordance with the ethical standards of the Institutional Review Board. Nineteen volunteer participants (age 25±3 years, 17 right-handed, all university students without medical conditions), were recruited after providing written informed consent.

Fig 1A depicts the experimental setup. The participants held with their right hand onto the handle of a planar robotic interface (KINARM, BKIN Technologies Inc., Ontario, Canada) to control its position, which was linked in real-time with ~1:1 proportion to a visualized cursor

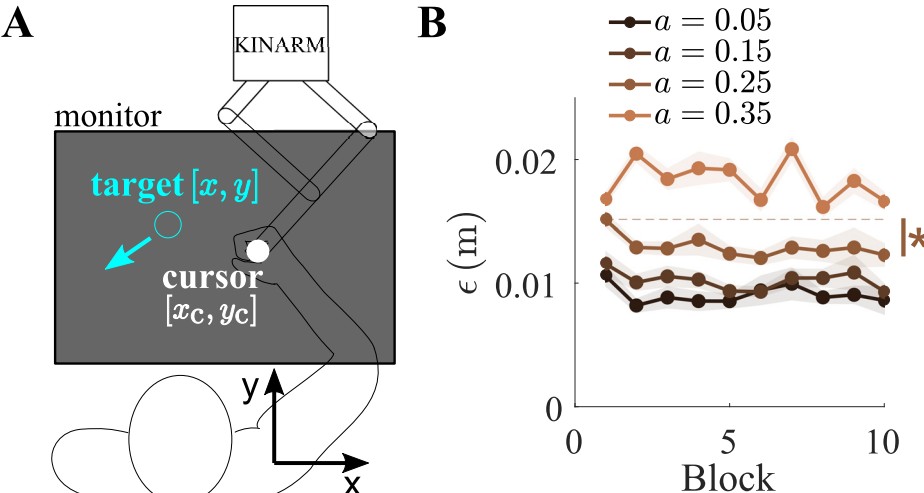

**Fig 1. Experimental setup and tracking error.** (A) Participants held onto the handle of the robotic interface (white cursor) to track the planar motion of a target (red). (B) Group mean cumulative tracking error $\epsilon$ as a function of block number, separately for each setting of $a$. Shaded areas denote standard error, and * indicates $p < 0.05$. Only significant comparisons are shown.

[21]. They viewed the cursor on a horizontal computer monitor that shielded the hand and arm from direct sight. This robot has an elliptical workspace of 0.76×0.44 m, is capable of generating a peak force pulse up to 58 N, and is equipped with a multiaxial force sensor. A moving rest was provided to ensure properly planar arm movement. During the tracking task, the following kinematic parameters were recorded: the planar position of the hand $[x_c, y_c]$, its velocity, the force exerted $\mathbf{F}$ and the grasp force $G$.

In addition, the EMG activity from nine muscles (namely, wrist muscle pair, elbow monoarticular pair, biarticular pair and shoulder muscles) was acquired via wireless sensors (picoEMG, Cometa S.r.l., Bareggio MI, Italy). The EEG was digitized using a 64-channel system (ActiveTwo, BioSemi, Amsterdam, Netherlands). All participants rested their chin and forehead on a headrest and the cable was secured to attenuate movement artifacts. Eight frontal electrodes were excluded due to headrest contact. The respiratory activity was monitored, separately for the thorax and abdomen compartments, using pneumatic sensor belts [22]. The plethysmographic signal, indexing cardiovascular arousal, was recorded via a photoplethysmograph (type 8600; Nonin Medical Inc., Plymouth, MN, USA). All data were digitized at 1000 Hz, with the exception of the EEG, which was recorded at 2048 Hz.

### Task design

The planar target position $[x, y]$ at time $t$ (omitted for brevity) was governed by the rescaled Rössler equations [23], namely,

$$\begin{cases} \dot{x} &= \omega(-y - z) \\ \dot{y} &= \omega(x + ay) \\ \dot{z} &= \omega(b + z(x - c)) \end{cases} \tag{1}$$

wherein we set $b = 0.2$, $c = 5.7$, $\omega = 3$ and the initial conditions were set to $y = z = 0$ and $x \in [6, 7]$, drawn randomly for each trial. The control parameter $a$ was varied to determine the level of chaoticity and folding (effectively, trajectory irregularity), over $a = \{0.05, 0.15, 0.25, 0.35\}$, wherein $a = 0.05$ represents a periodic circular orbit and $a = 0.35$ corresponds to fully-

developed chaoticity; these properties are well-established and discussed, for example, in Refs. [2, 24–28]. The chaotic nature, visible as irregular fluctuations in the peak amplitudes and cycle durations, is well-evident on the $(x, y)$ plane as the initially closed circular trajectory (limit cycle) is replaced by a dense superposition of non-overlapping orbits that become increasingly folded and visit an increasing proportion of the bounded area (Fig 2A). For the avoidance of doubt, it should be pointed out that chaotic dynamics are well-evident even when the $z$ variable of the system is disregarded, as implied by Takens' theorem, which allows for reconstructing an attractor from time-lag embedding based on a single variable [29, 30]. On this basis, the largest Lyapunov exponent $\lambda_{MAX}$ and the correlation dimension $D_2$ can be readily calculated even from the separate $x$ and $y$ time-series. As documented in Table 1, for $a = 0.05$, one has $\lambda_{MAX} < 0$ and $D_2 \approx 1$, indicating periodic dynamics; for $a \geq 0.15$, both measures monotonically increase until $\lambda_{MAX} \approx 0.07$ and $D_2 \approx 2$, hallmarking the low-dimensional chaotic dynamics that knowingly characterize this attractor [31–33]. Accordingly, the autocorrelation functions, which initially oscillate between ±1, decay faster with increasing $a$, representing the loss of periodicity (Fig 3).

The outputs $[x, y]$ were multiplied by a scaling factor of 0.005 to yield the target coordinates in meters. The integration time $t$ was set to correspond to physical time in seconds. The system was integrated in fixed steps of 0.0005s using the Runge-Kutta order 4 method implemented in real-time (Simulink, MathWorks Inc., Natick MA, USA).

To assess the force exerted by each participant during tracking, the robot imposed a friction

$$\mathbf{F}_r = -\mu \begin{bmatrix} \dot{x}_c \\ \dot{y}_c \end{bmatrix} \tag{2}$$

where the viscous friction coefficient was set to $\mu = 30$Ns/m, appreciably opposing hand motion.

The experiment totaled 40 trials, each having a duration of 45 s and corresponding to approximately 21 periods in the limit-cycle case. Participants experienced the trials in 10 blocks, wherein each block contained trials with the control parameter randomly selected from the four levels. A 15 s rest preceded each trial to reduce fatigue.

## Results

### Tracking accuracy

We firstly examined how the tracking error depended on the control parameter setting $a$. Representative target and cursor trajectories can be viewed in Fig 2A. The cumulative tracking error was defined as

$$\epsilon = \frac{1}{T} \int_{t=0}^{T} \sqrt{(x - x_c)^2 + (y - y_c)^2}\, dt. \tag{3}$$

Fig 1B shows the group mean cumulative tracking error $\epsilon$ as a function of block number, separately for each setting of $a$. To assess the influence of the block number, we calculated the difference in $\epsilon$ between the first and last block, separately for each control parameter setting. A two-way repeated-measures ANOVA revealed that both the control parameter ($p < 0.001$, $F(3, 54) = 62$) and the block number ($p = 0.02$, $F(1, 18) = 6.3$) had significant main effects. Planned comparisons (Tukey's HSD) showed that $\epsilon$ was different between all settings of $a$, but the difference in $\epsilon$ between the first and last block was only significant for $a = 0.25$. In the last block, $\epsilon$ = {0.0087±0.0012, 0.0093±0.0007, 0.0123±0.0009, 0.0166±0.0007}m (mean±standard error),

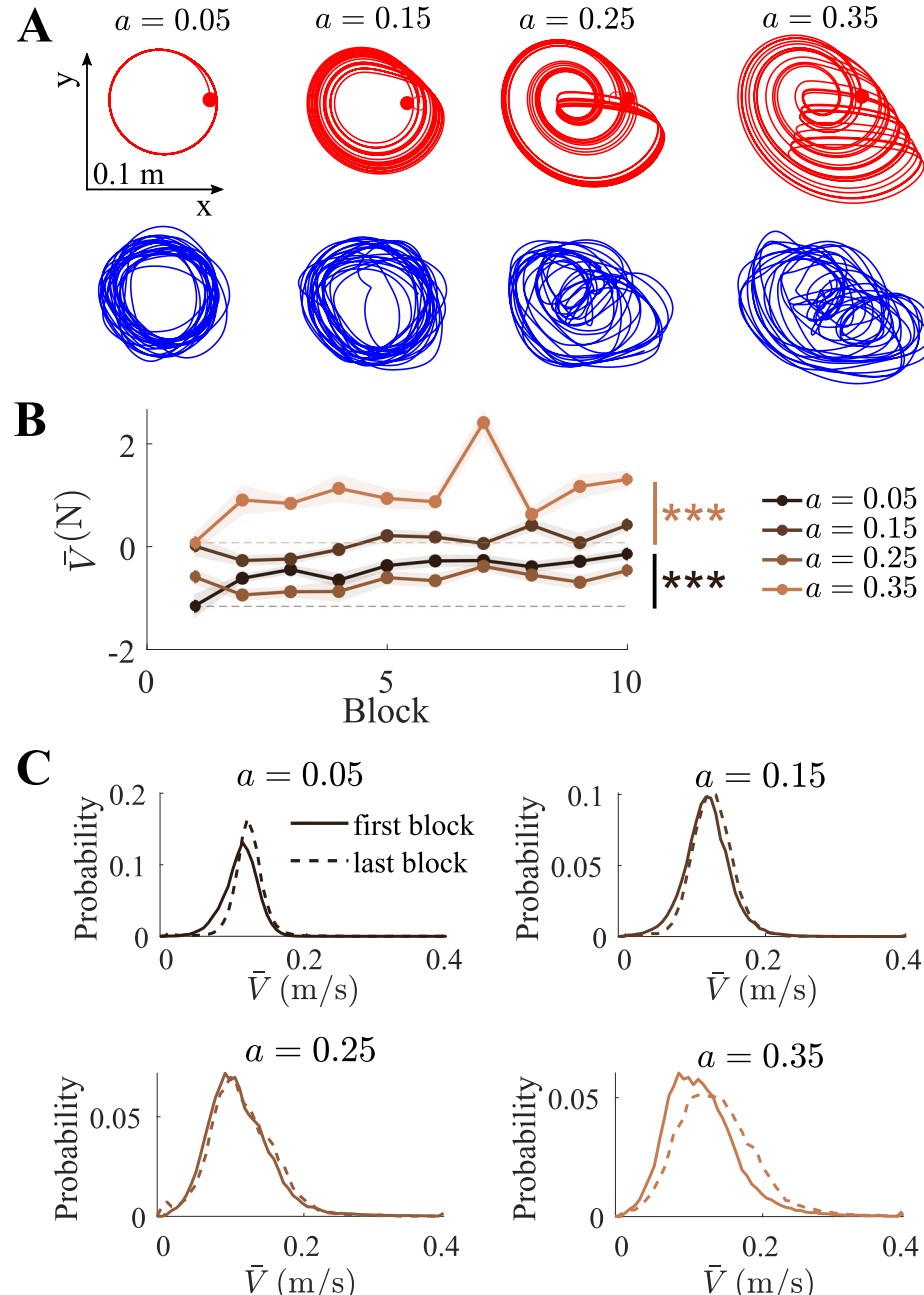

**Fig 2. Control parameter regulated the chaoticity in the target's motion.** (A) Representative trajectories of the target $[x, y]$ (red) and cursor $[x_c, y_c]$ (blue). As the control parameter setting $a$ was elevated, increased folding and irregularity became more evident, resulting in lower tracking accuracy. (B) Normalized cursor velocity magnitude $\bar{V}^{(N)}$ as a function of the block number. $\bar{V}^{(N)}$ tended to increase over time. (C) Probability density function $\bar{V}$ in the first and last blocks showed reduced incidence of low-velocity movements with practice. *** represents $p < 0.001$.

for increasing $a$. That is, tracking accuracy was lowest under the fully-developed chaoticity condition, and it improved over time under intermediate settings of the control parameter.

Movement is knowingly intermittent during tracking [34] as participants make correctional submovements to accurately track the target's position [35]. The number and duration of submovements are expected to decrease with practice [36], leading to an overall increase in the

**Table 1. Non-linear dynamical parameters as a function of the bifurcation parameter $a$, i.e., largest Lyapunov exponent $\lambda_{MAX}$ (step size: 0.025 s) and correlation dimension $D_2$ calculated for the scalar $x$ and $y$ coordinate time-series.**

| $a$ | $\lambda_{MAX}, x$ | $\lambda_{MAX}, y$ | $D_2, x$ | $D_2, y$ |
|---|---|---|---|---|
| 0.05 | -0.002 ± 0.002 | -0.002 ± 0.002 | 1.03 ± 0.02 | 1.06 ± 0.06 |
| 0.15 | 0.006 ± 0.001 | 0.003 ± 0.003 | 1.07 ± 0.16 | 1.08 ± 0.16 |
| 0.25 | 0.027 ± 0.003 | 0.028 ± 0.004 | 1.70 ± 0.08 | 1.72 ± 0.07 |
| 0.35 | 0.070 ± 0.007 | 0.072 ± 0.011 | 1.95 ± 0.08 | 1.95 ± 0.06 |

cursor's velocity magnitude. The average magnitude of the cursor's velocity $V = \sqrt{\dot{x}_c^2 + \dot{y}_c^2}$ was calculated in every trial to yield $\bar{V}$ (Fig 2B). For comparison, $\bar{V}$ was normalized by $z$-transformation within each participant over all trials; hereafter, for the avoidance of doubt normalized variables are denoted with superscript "(N)". A two-way repeated measures ANOVA revealed a significant effect of the control parameter setting ($p < 0.001$, $F(3, 54) = 39$) and the block number ($p = 0.002$, $F(1, 18) = 15$) on $\bar{V}^{(N)}$. Planned comparisons showed that $\bar{V}$ increased over time for $a = \{0.05, 0.35\}$. The probability density function of $\bar{V}$ in the first and last blocks showed a reduction in the incidence of low velocity movements, which was noticeable for $a = \{0.15, 0.25\}$ and significant for $a = \{0.05, 0.35\}$ (Fig 2C), suggesting that the sub-movements may have subsided with practice.

## Learning to predict the fold

The error $\epsilon$ crudely quantifies the mean distance between the target and cursor positions over entire trials. To focus on the possible learning of the chaotic dynamics, we next examined more finely how the participants reacted to the folds in the target trajectory. As visible in Fig 4A, for high levels of $a$, the folding was characterized by sharp transients in the target acceleration magnitude $A = \sqrt{\ddot{x}^2 + \ddot{y}^2}$; therefore, we calculated $\bar{A}$, the average target acceleration magnitude, for every trial. Accordingly, a one-way repeated measures ANOVA revealed a significant effect of the control parameter on $\bar{A}^{(N)}$ ($p < 0.001$, $F(3, 54) = 16000$), and all post-hoc

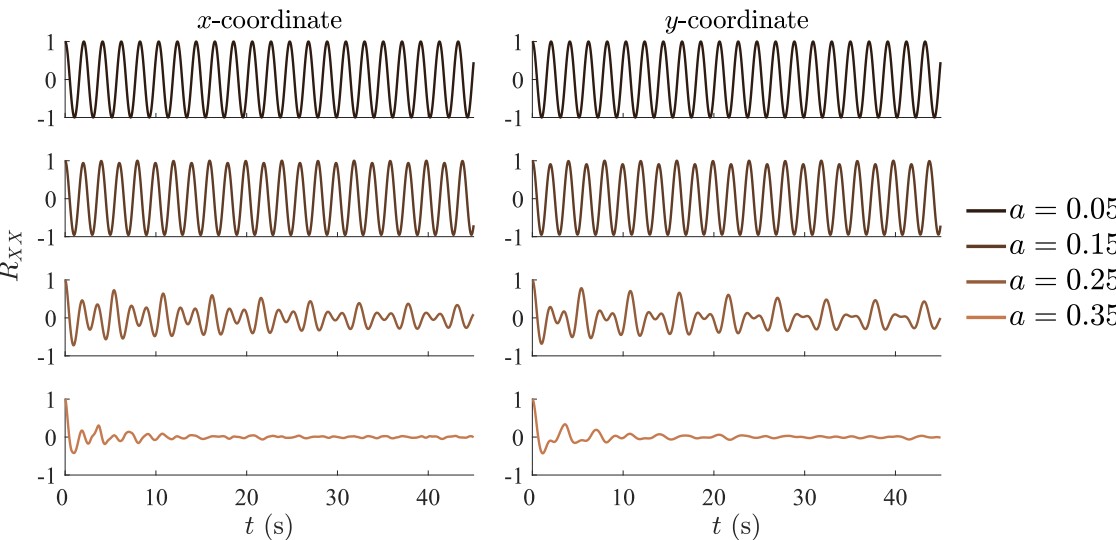

**Fig 3. Autocorrelation functions for the target trajectory coordinates.** The autocorrelation along $x$ and $y$ initially oscillates around ±1, decaying faster with increasing control parameter $a$, representing the loss of periodicity.

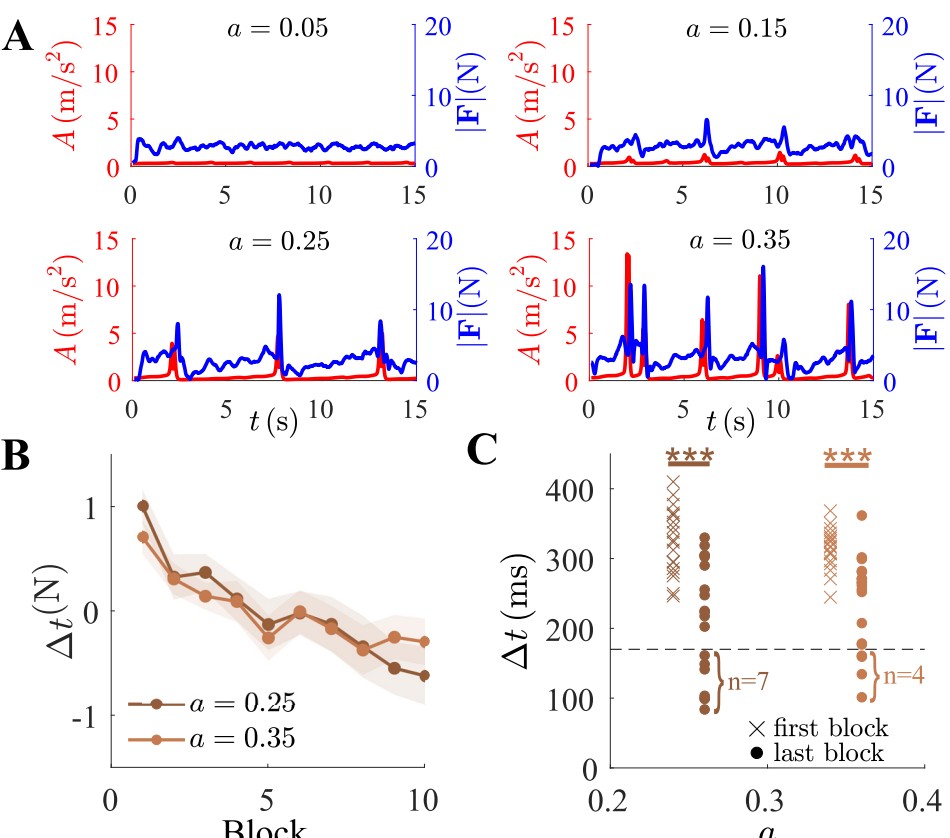

**Fig 4. Anticipation of the transient target occurred with practice.** (A) The chaotic target's acceleration magnitude $A$ (red) and the force magnitude $|\mathbf{F}|$ (blue) as a function of time from four sample trials with increasing control parameter $a$ from top-left to bottom-right. (B) Normalized movement delay $\Delta t^{(N)}$ as a function of the block number. (C) $\Delta t$ in the first and last blocks as a function of control parameter $a$ for each individual participant. With practice, the participants reduced their movement delay in response to the fold. *** represents $p < 0.001$.

comparisons were significant, confirming that the average acceleration increased with chaoticity. In response to the target transients, corresponding peaks in force magnitude $|\mathbf{F}|$ were observed. These were initially reactive, temporally lagging behind the target trajectory. To quantify this lag, we defined the peak-to-peak time $\Delta t$ between the acceleration and force magnitude, $A$ and $|\mathbf{F}|$. The normalized lag $\Delta t^{(N)}$ was only calculated for $a = \{0.25, 0.35\}$ because folds are not generated at the lower settings.

The group mean normalized $\Delta t^{(N)}$ markedly decreased as a function of the block number (Fig 4B). A two-way repeated measures ANOVA confirmed that, while it was comparable between the two chaotic settings of the control parameter ($p = 0.7$), it significantly decreased over time ($p < 0.001$, $F(1, 18) = 38$). During a tracking task, $\approx170$ms are knowingly needed for humans to initiate movement in response to a target event [34, 37]. Accordingly, no participant reacted to the fold faster than this during the first block (Fig 4C). However, by the last block, 7 participants for $a = 0.25$ and 4 participants for $a = 0.35$ had attained a $\Delta t < 170$ms. This suggests that 37% and 21% of them no longer reacted to the occurrence of the fold, but may have learned to anticipate it.

We also explicitly considered the degenerate possibility that the participants could be trivially tracking the phase of the limit cycle orbit, that is, ignoring the fold and following a circular motion. If so, averaging out the time-dependence of amplitude (i.e., distance from the origin)

should have no effect on tracking accuracy. By comparing the behavioral tracking error $\epsilon$ with a surrogate error $\epsilon_s$, we probed more stringently whether the participants tracked the target's position during the fold. Namely, given $\langle x \rangle = \langle y \rangle = 0$, the surrogate error $\epsilon_s$ was

$$\epsilon_s = \frac{1}{T} \int_0^T \sqrt{(x - x_s)^2 + (y - y_s)^2} \, dt \tag{4}$$

with the surrogate cursor position $[x_s, y_s]$ given by

$$\begin{aligned} x_s &= \mathrm{Re}(\langle A \rangle e^{i\psi}) \\ y_s &= \mathrm{Im}(\langle A \rangle e^{i\psi}) \end{aligned} \tag{5}$$

where $A$ and $\psi$ denote the amplitude and phase of the complex-valued cursor position $x_c + iy_c$. We calculated the difference between the surrogate error and the error, $\Delta\epsilon = \epsilon_s - \epsilon$, for all trials. A one-way repeated measures ANOVA revealed a significant effect of the control parameter setting on $\Delta\epsilon$ ($p < 0.001$, $F(3, 54) = 863$). All post-hoc comparisons were significant, except between $a = \{0.25, 0.35\}$.

The signed difference $\Delta\epsilon$ provides additional information about the tracking accuracy. When $a = 0.05$ and the target's motion was circular, $\epsilon_s < \epsilon$ (one-sample t-test, $t(18) = -15.7$, $p < 0.001$), implying that destroying the cursor's amplitude information improved tracking accuracy, plausibly due to a reduction in involuntary movement variability. For $a = 0.15$, $\Delta\epsilon$ was not significantly different from zero. For $a = \{0.25, 0.35\}$, the surrogate error $\epsilon_s > \epsilon$ ($t(18) = 17.2$, $p < 0.001$, and $t(18) = 15.4$, $p < 0.001$). Thus, the amplitude information was of critical importance during the trials with chaotic dynamics, suggesting that the participants were effectively tracking the folding orbit.

## Adaptation of the exerted force

To examine the change in the magnitude of the force applied by each participant, we calculated

$$\bar{F} = \frac{1}{T} \int_0^T |\mathbf{F}| \, dt \, . \tag{6}$$

A two-way repeated measures ANOVA indicated that both the control parameter setting ($p < 0.001$, $F(3, 54) = 29$) and the block number ($p = 0.004$, $F(1, 18) = 11$) had a significant effect on the normalized force $\bar{F}^{(N)}$. Namely, $\bar{F}^{(N)}$ increased significantly for $a = \{0.05, 0.35\}$, while it remained constant in other settings. The increase in $\bar{F}^{(N)}$ mirrors the increase in the cursor's velocity magnitude $\bar{V}$ (Fig 2B).

Motor learning of a task generally results in the reduction of the position feedback gain [38], which can be estimated from the force $\mathbf{F}$ through a spring-like linear control model [39]. $\mathbf{F}$ has a velocity-dependent component due to the robot's viscous friction opposing the participant's motion (Eq 2), which must be removed from $\mathbf{F}$ to estimate the linear control model. The viscous force was approximated by

$$\mathbf{F} \approx L_v \begin{bmatrix} \dot{x}_c \\ \dot{y}_c \end{bmatrix} , \tag{7}$$

where the viscous gain $L_v$ is assumed to be constant. $L_v$ was calculated in every trial using least-squares regression, yielding $R^2 \approx 0.96$. As expected, the group mean was $L_v = 32.9 \pm 0.6 \mathrm{Ns/m} \approx \mu$. A two-way repeated measures ANOVA revealed a significant influence of the control

parameter setting ($p < 0.001$, $F(3, 54) = 22$) and the block number ($p = 0.02$, $F(1, 18) = 7$) on $L_v^{(N)}$. In order of increasing $a$, in the first block $L_v = \{33.5\pm0.6, 32.4\pm0.9, 33.0\pm0.6, 32.9\pm0.6\}$ Ns/m, and in the last block it decreased to $L_v = \{31.9\pm1.0, 32.6\pm0.6, 32.5\pm0.7, 32.3\pm0.6\}$Ns/m, converging towards $\mu$.

We thereafter estimated the linear control model by approximating the residual force as a function of the target and cursor's positions and velocities according to

$$\mathbf{F} - L_v \begin{bmatrix} \dot{x}_c \\ \dot{y}_c \end{bmatrix} = \Delta\mathbf{F} \approx L_p \begin{bmatrix} x - x_c \\ y - y_c \end{bmatrix} + L_d \begin{bmatrix} \dot{x} - \dot{x}_c \\ \dot{y} - \dot{y}_c \end{bmatrix} \tag{8}$$

where the position feedback gain $L_p$ and velocity feedback gain $L_d$ are assumed to be constant. The model assuming the force is determined by a spring and damper yielded an F-value $\approx 0.6$ ($p = 0.6$). This could not be improved by introducing quadratic and cross-terms, apparently excluding the possibility of a non-linear dependence. A model with the cursor's acceleration

$$\Delta\mathbf{F} \approx I \begin{bmatrix} \ddot{x}_c \\ \ddot{y}_c \end{bmatrix} \tag{9}$$

where $I$ is a constant, yielded a marginally better model with a higher F-value $\approx 1.9$ ($p = 0.2$), suggesting that $\Delta\mathbf{F}$ was, albeit weakly, more closely dependent on the arm's inertia. As the linear control models in Eqs 8 and 9 cannot sufficiently explain the variation in $\Delta\mathbf{F}$, a more sophisticated control model is needed in the future to approximate the control law that emerges when tracking a chaotic target.

## Reduction of muscular co-contraction and grasp force

The human arm features multiple joints, the motion of each being controlled by an agonist-antagonist muscle pair. As a means of adapting to task conditions, such as performing fine movements, a muscle pair can co-activate or co-contract, resulting in zero net torque (no force) but increased joint stiffness [40]. The magnitude of the arm's endpoint stiffness is also often positively related to the grasp force $G$[41, 42]. Consequently, a reduction in the muscular co-contraction and the grasp force is typically observed while learning a model of the dynamics in a new task [15, 16, 43].

To investigate this possibility, the raw EMG activity from each muscle was firstly high-pass filtered (second-order Butterworth filter at >10Hz), rectified, then low-pass filtered at <3Hz, yielding positive-valued filtered voltage time-series $m_i$ for the nine arm muscles $i = 1...9$. Muscle co-contraction $u$ was thereafter empirically estimated from the average activity in the entire arm [44], assuming

$$u = \frac{1}{9T} \sum_{i=1}^{9} \int_0^T m_i \, dt \, . \tag{10}$$

We did not analyze the activity of each muscle individually, as it was highly correlated within each block ($r \approx 0.96$).

A two-way repeated measures ANOVA showed that the normalized co-contraction measure $u^{(N)}$ was dependent on both the control parameter setting ($p = 0.01$, $F(3, 54) = 3.9$) and the block number ($p < 0.001$, $F(1, 18) = 35$). Planned comparisons revealed that it decreased significantly over time (Fig 5A), and was greater for $a = 0.35$ than $a = 0.15$.

Rhyming with this finding, a noticeable decrease in the normalized grasp force $G^{(N)}$ also occurred over time (Fig 5B). A two-way repeated measures ANOVA revealed a significant

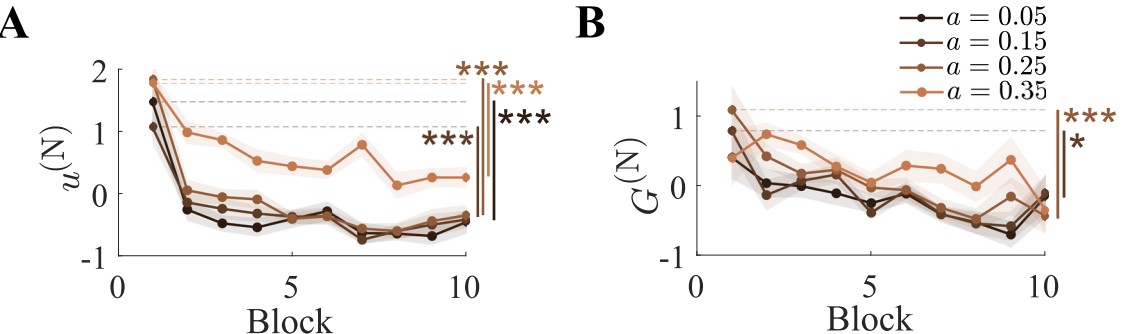

**Fig 5. Co-contraction and the grasp force decreased with the block number.** (A) Normalized muscle co-contraction $u^{(N)}$ and (B) grasp force $G^{(N)}$ as a function of the block number for each control parameter setting $a$. Both $u^{(N)}$ and $G^{(N)}$ decreased over time. $*$ represents $p < 0.05$ and $***$ represents $p < 0.001$.

effect of the block number on $G^{(N)}$ ($p = 0.05$, $F(1, 18) = 4.6$), with the control parameter setting exerting no influence ($p = 0.5$). Planned comparisons showed that $G^{(N)}$ decreased significantly for $a = \{0.15, 0.25\}$. Together with the decrease in co-contraction, this illustrates the reduction in arm stiffness due to training.

## Autonomic entrainment

We next examined whether the tracking task influenced bodily arousal as indexed by cardiore-spiratory physiology and as previously observed, for instance, in response to economic parameters during decision-making [45]. A one-way repeated measures ANOVA revealed that neither the breathing rate ($p = 0.6$), nor the plethysmogram amplitude ($p = 0.8$), nor the heart rate ($p = 0.6$) were influenced by the chaoticity level, suggesting that even the most taxing setting of the control parameter did not engender significant autonomic activation.

A finer-grained analysis was then performed to probe the possible synchrony between task-related movement and breathing, as measured in the thorax $b_{th}$ and abdomen $b_{ab}$: this evaluation was particularly motivated by the prior knowledge that volitional rhythmic movements engender synchronization in respiration [18–20]. A representative side-by-side comparison of the breathing signals, representing an approximation of tidal volume, and a component of the cursor motion is visible in Fig 6. We calculated the phase locking between $b_{th}$ and $b_{ab}$, on the one hand, and $x_c$ and $y_c$ on the other. For each time-series $s = \{b_{th}, b_{ab}, x_c, y_c\}$, the corresponding analytic signal was calculated as

$$\psi = s + i\tilde{s} = Ae^{i\phi}, \tag{11}$$

where $i = \sqrt{-1}$ and $\tilde{s}$ denotes the Hilbert transform of $s$

$$\tilde{s} = \frac{1}{\pi}\text{p.v}\int_{-\infty}^{\infty}\frac{s}{t - \tau}d\tau, \tag{12}$$

where p.v represents the Cauchy principal value of the integral. From these, the instantaneous phases $\phi_{th}$, $\phi_{ab}$, $\phi_{x_c}$ and $\phi_{y_c}$ were obtained.

The phase synchronization between the respiratory compartment expansions, alas $\phi_{th}$ and $\phi_{ab}$, and the cursor coordinates $\phi_{x_c}$ and $\phi_{y_c}$ was computed, yielding four values of the phase synchronization index $S$. The synchronization between $\phi_{th}$ and $\phi_{x_c}$ was assessed as

$$S_{th,x_c} = \frac{1}{T}\left|\int_0^T e^{i(\phi_{th} - n\phi_{x_c})}\,dt\right|, \tag{13}$$

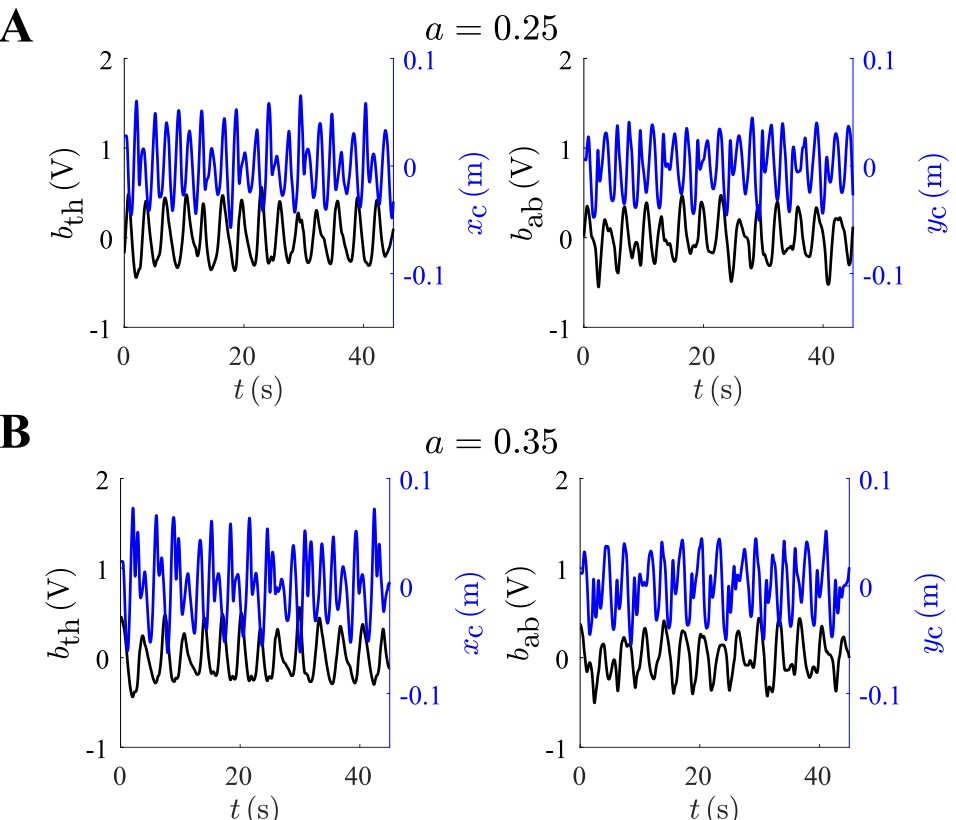

**Fig 6. Expansion of the thorax and the abdomen from a representative participant.** Time-series for expansion of the thorax $b_{th}$ vs. cursor's position $x_c$ (left), and the abdomen expansion $b_{ab}$ vs. $y_c$ (right). (A) Representative trial given $a = 0.25$, and (B) trial with $a = 0.35$, from the same participant in Fig 2A.

where $n$ is a frequency ratio, and similarly for the other combinations. These values were then averaged as $S_{th} = \frac{1}{2}S_{th,x_c} + \frac{1}{2}S_{th,y_c}$ and $S_{ab} = \frac{1}{2}S_{ab,x_c} + \frac{1}{2}S_{ab,y_c}$; here, the two compartments were treated separately to confirm that the effect was unlikely to stem from a motion artefact. In addition to $n = 1$, we considered the frequency ratios $n = \left[\frac{1}{2}, 2\right]$, averaged across all blocks and control parameter settings. In order of increasing $n$, $S_{th} = \{0.13 \pm 0.01, 0.15 \pm 0.01, 0.04 \pm 0.01\}$ and $S_{ab} = \{0.20 \pm 0.03, 0.28 \pm 0.04, 0.04 \pm 0.01\}$, demonstrating that the synchronization was strongest assuming a unitary frequency ratio, corresponding to one breathing cycle per orbit period.

In Fig 7A, the normalized synchronization indices $S_{th}^{(N)}$ and $S_{ab}^{(N)}$ for $n = \left[\frac{1}{2}, 1, 2\right]$ are charted as a function of the control parameter setting. A two-way repeated measures ANOVA revealed no effect of the frequency ratio but a significant effect of the control parameter setting on both $S_{th}^{(N)}$ ($p < 0.001$, $F(3, 54) = 39$) and $S_{ab}^{(N)}$ ($p < 0.001$, $F(3, 54) = 6.7$). Planned comparisons confirmed significant synchronization differences in the thorax between $a = \{0.05, 0.15, 0.25\}$ and $a = 0.35$ for $n = \left[\frac{1}{2}, 1, 2\right]$, and between $a = \{0.15, 0.25\}$ for $n = 2$ alone. Significant differences in the abdomen were found for the double frequency ratio between $a = \{0.05, 0.15, 0.25\}$ and $a = 0.35$, and between $a = \{0.05, 0.15, 0.35\}$ and $a = 0.25$. A significant difference was also found between $a = \{0.15, 0.25\}$ and $a = 0.35$ for the half frequency ratio. Thus, markedly greater entrainment emerged when tracking a target with a chaotic orbit. This effect could be similarly discerned on the scale of identity, half and double frequency ratios.

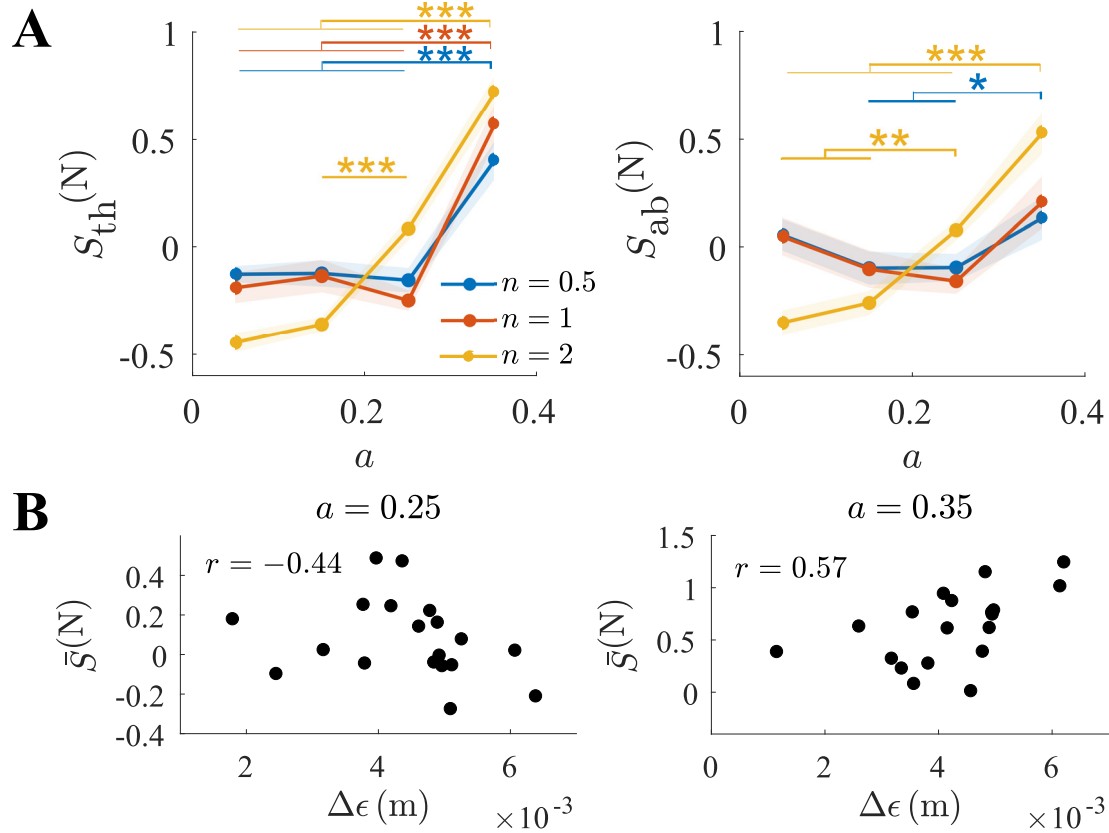

**Fig 7. Synchronization between the breathing and the cursor's motion was strongest during fully-developed chaos.** (A) Normalized synchronization indices $S_{th}^{(N)}$ and $S_{ab}^{(N)}$ as a function of the control parameter $a$, shown for different values of the frequency ratio $n$ (period doubling and halving). (B) Normalized $\bar{S}^{(N)}$ as a function of $\Delta\epsilon$ for $a = \{0.25, 0.35\}$. Borderline and significant rank correlation were observed for $a = \{0.25, 0.35\}$, respectively. * represents $p < 0.05$, ** represents $p < 0.01$, and *** represents $p < 0.001$.

We lastly speculated that breathing and entrainment could be functional, or otherwise related, to tracking accuracy. More specifically, we postulated that the emergent synchronization, which was strongest under the partially and fully-developed chaos conditions, was related to the tracking of the fold. To evaluate this possibility, we examined Spearman's correlation coefficient between the error difference $\Delta\epsilon$ and the average synchrony of the thorax and abdomen $\bar{S}^{(N)} = \frac{1}{2}S_{th}^{(N)} + \frac{1}{2}S_{ab}^{(N)}$ for $a = \{0.25, 0.35\}$ (Fig 7B). The rank-order correlation was borderline not significant for $a = 0.25$ ($r = -0.44$, $p = 0.06$), but $\bar{S}^{(N)}$ and $\Delta\epsilon$ were significantly positively correlated for $a = 0.35$ ($r = 0.57$, $p = 0.01$). The different signs possibly reflected boosted performance with more intense synchronization under the intermediate chaoticity setting, and increased but ineffective effort with more intense synchronization under the highest chaoticity setting.

### Modulation of neuroelectrical activity

To gain further insight, albeit at a coarse-grained level, into the neural correlates of task performance, we conducted a topographical power analysis on the EEG rhythms across the $\delta$ ([1, 4] Hz), $\theta$ ([4, 8] Hz), $\alpha$ ([8, 15] Hz) and $\beta$ ([15, 32] Hz) bands. For each, the power was quantified

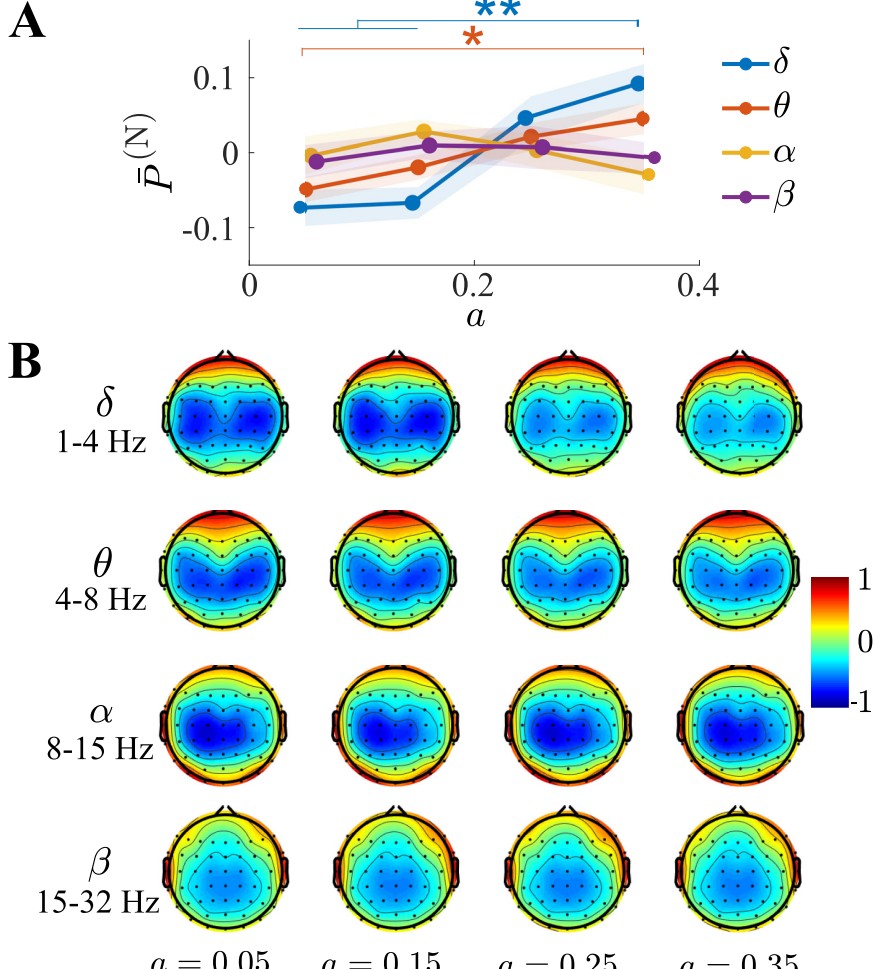

**Fig 8. EEG rhythms in the $\theta$ and $\delta$ bands increased with the control parameter setting.** (A) Normalized EEG power $\bar{P}^{(N)}$ as a function of the control parameter setting, separately for the $\delta$, $\theta$, $\alpha$ and $\beta$ bands. Both $\theta$ and $\delta$ activity increased as a function of $a$, whereas $\alpha$ and $\beta$ did not. (B) Topographical plots of normalized EEG power. Larger oscillations were observed over the frontal region in the $\delta$ and $\theta$ bands. $^*$ represents $p < 0.05$ and $^{**}$ represents $p < 0.01$.

over all channels and trials, then separately normalized based on the channel mean, and finally averaged as $\bar{P}^{(N)}$.

We first considered how $\bar{P}^{(N)}$ in each band depended on the control parameter setting. As charted in Fig 8A, separate one-way ANOVAs with the control parameter as factor revealed that it significantly influenced $\bar{P}^{(N)}$ for the $\delta$ ($p < 0.001$, $F(3, 54) = 8.1$) and $\theta$ ($p = 0.01$, $F(3, 54) = 4.0$) bands, but not the $\alpha$ ($p = 0.5$) and $\beta$ bands ($p = 0.9$). Post-hoc comparisons in the $\delta$ band were significant throughout, except between $a = \{0.05, 0.15\}$, and between $a = \{0.25, 0.35\}$. Post-hoc comparisons in the $\theta$ band were significant only between $a = \{0.05, 0.35\}$. These results altogether point to increased generation of coherent oscillations in the $\delta$ and $\theta$ bands with increasing target chaoticity.

The distribution of $\bar{P}^{(N)}$ over the scalp is visible through the topographical plots in Fig 8B, generated using the EEGLAB software [46]. With ensuing chaoticity, larger oscillations emerged over the frontal and central regions in both the $\delta$ and $\theta$ bands, which, altogether, suggest an enhanced activity related to motor imagery and execution [47, 48].

## Discussion

In line with the existing literature [11], the present results point to a remarkable innate ability of tracking a chaotically-moving target plausibly based on the underlying topological regularities, in this case the location of the fold, despite the inherent unpredictability and complexity of the trajectory. Overall, tracking accuracy predictably declined with increasing chaoticity, however, intermediate settings opened way to a learning effect whereby the occurrence of the acceleration transients was eventually anticipated. At the lowest settings, this effect was unobservable due to the lack of folding, whereas at the highest setting its emergence was probably hindered by excessive irregularity.

The cycle-level reactions to the fold were well-evident in the force magnitude time-series. The time delay known to be required for a reaction to a target event is 170ms [34, 37]. Initially, none of the participants acted faster than that, however, up to 37% of them eventually anticipated the fold by updating their trajectory faster than that. It appears plausible that the fold could be predicted at first through memorizing the probability distribution of its location in the task workspace, corresponding to a particular region of the bidimensional projection of the phase space, approximately mapped to the ($x > 0$, $y < 0$) quadrant (Fig 2A).

The nervous system does not learn novel dynamics through rote memorization [14, 49], but does so by acquiring a representation of it [50], which would correspond to a model of the chaotic dynamics in our task. Our results, however, do not yet shed any light on the possible structure and properties of such a model. Numerically, nonlinear time-series prediction is often attained by means of low-order predictors based on the neighbour points in a suitably high-dimensional embedding space [33]. However, due to its highly abstract nature, it is implausible that the brain approximates such a representation. More probably, participants may have developed a simple heuristic based on statistical considerations of the likely area of occurence of the fold, together with implicit learning of subtle cues, such as increased curvature away from the limit cycle orbit, or other fluctuations supporting the prediction of an impending transient. In this sense, a limitation is that the present study does not fully differentiate between proper, anticipatory prediction and ability to track a posteriori with increasing reactiveness.

Generally, the learning of a model is associated with a reduction in the position feedback gain, reflecting the fact that participants can use the force more resourcefully while achieving constant or even improved tracking accuracy [38]. Most of the force could be explained by the viscous friction opposing the participant's motion, but the residual force could not be explained by a linear control model consisting of a spring and a damper [39, 51], nor was it improved by adding quadratic or cross-terms. The failure in our model could, in part, be due to the assumption that participants track the target's accurate state. Its state is estimated by using delayed and noisy visual information, resulting in estimation errors [52]. Our model would fail to explain the exerted force if a faulty estimate of the target is tracked. A more sophisticated model may attempt to estimate the participant's faulty target state, but no such algorithm or method exists to date, thereby limiting our ability to model the control law when tracking a chaotic target.

Another indication that the participants could have learned a model of the chaotic dynamics, in the form of associating the target's current state to anticipate its future state, was obtained from the reduction of muscular co-contraction and grasp force observed over time [15, 16, 43]. Despite the magnitude of the force increasing with time, owing to an increase in the cursor's velocity magnitude, participants managed to decrease their overall muscle activity with practice. Additionally, a decrease in grasp force, which is positively correlated with the arm's endpoint stiffness magnitude [42], was observed. In this regard, it should be

acknowledged that even though decreased co-contraction and grasp force usually accompany motor learning, a more parsimonious explanation could be that of simple parametric adaptation to the task requirements, without any underlying model formation process.

The changes in brain activity observed under the different levels of chaoticity lend further support to the view that the participants may have acquired a model of the dynamics, reflected in the selection of different strategies and consequently brain states. Higher chaoticity engendered an increase in neural activity over the $\delta$ and $\theta$ frequency bands. Typically, elevated power in these bands is observed when processing errors [53], at the onset of motor imagery [47, 54] and during motor execution [48, 55–57]. Further, a marked increase in $\delta$ and $\theta$ band power is observed during motor imagery across the thalamus, cortex and cerebellum [47]. The cerebellum also plays an important role in the prediction of the future states [13, 58], and is critical to a tracking task where the hand must be moved to the anticipated target's position using delayed visual feedback [59].

The initiation of voluntary actions is oftentimes associated with exhalation during respiration [20, 60]. Based on the canonical responses to stress, one could expect faster respiration and heart rate under higher chaoticity, but no significant differences in cardiovascular activation were observed. Yet, markedly elevated synchronization between respiration and the tracking movement emerged with increasing levels of chaoticity. The synchronization between respiration and voluntary motion is a well-established and pervasive phenomenon, especially when the movement is cyclical such as during locomotion [18, 61] or otherwise rhythmic [19, 62, 63]. To our knowledge, however, the degree of synchronization had not been tied with the difficulty of a motor task. A larger control parameter setting was also related to greater neural activity in the $\theta$ band, which in turn can reflect sustained attention and focus, e.g., during meditation [17, 64]. Thus, the most plausible explanation is that under the more difficult conditions of larger control parameter setting, increased mental effort and focus drove the stronger respiration synchronization. Based on this experiment, it is not possible to finally ascertain whether the correlation with tracking error was epiphenomenal or functional to performance. Contamination due to movement artefacts also cannot be fully excluded, though it appears highly unlikely due to the coherent effect on compartments probed at anatomically well-separated locations (nipple and umbilicus levels).

Several studies have examined the human ability to predict chaotic sequences [8–10], but to our knowledge only one study had examined the prediction of chaotic dynamics using a motor task [11]. The study examined the effect of increasing feedback delay on the cursor's position when tracking a target's motion governed by a chaotic spring, whose stiffness was controlled by a Röessler system. The visual feedback delay was manipulated to examine its influence on the correlation between the cursor and the target trajectory, whose motion consists of smooth elliptical orbits. Beyond this correlational analysis, the authors did not delve into kinematic measures such as tracking error, and could not analyze changes in muscle activity or force as only the cursor's position and velocity were measured. In contrast to the present work, the parameters of the Rössler system in Ref. [11] ($a = b = 0.1$, $c = 14$) did not generate transients in the target position, but rather more gentle cycle amplitude fluctuations. To the authors' knowledge, then, this work is the first to consider the Rössler system's ability to gradually control the level of chaoticity and the expression of a fold in a particular region of the phase space, thus offering a well-defined feature against which to measure motor performance and learning. Our study is also unique in its examination of the kinematics (tracking error and feedback gain) and the physiology (cardiorespiratory system, muscle and neural activity).

In summary, a significant body of evidence exists on the human ability to anticipate ballistic trajectories or stationary dynamics [14, 65], and our results indicate that, to some extent, this ability extends to nonlinear transient dynamics. Our study is limited in the sense that we could

neither identify the information necessary to anticipate the fold, nor elucidate how it is represented by the brain. In future work, we plan on hiding sections of the target trajectory prior to the fold to examine the quantity of information required to anticipate it, and uncovering more precisely the control law which emerges in the presence of chaos. Additionally, the use of functional magnetic resonance imaging may provide clues to the regions involved in the learning of chaotic dynamics and their interactions.

## Author Contributions

**Conceptualization:** Atsushi Takagi, Ludovico Minati.

**Data curation:** Atsushi Takagi, Ryoga Furuta, Supat Saetia, Ludovico Minati.

**Formal analysis:** Atsushi Takagi, Ryoga Furuta, Ludovico Minati.

**Funding acquisition:** Atsushi Takagi, Natsue Yoshimura, Yasuharu Koike, Ludovico Minati.

**Investigation:** Atsushi Takagi, Ludovico Minati.

**Methodology:** Atsushi Takagi, Ludovico Minati.

**Software:** Atsushi Takagi, Ludovico Minati.

**Supervision:** Supat Saetia, Natsue Yoshimura, Yasuharu Koike, Ludovico Minati.

**Validation:** Atsushi Takagi, Ludovico Minati.

**Visualization:** Ludovico Minati.

**Writing – original draft:** Atsushi Takagi, Ludovico Minati.

**Writing – review & editing:** Atsushi Takagi, Ryoga Furuta, Supat Saetia, Natsue Yoshimura, Yasuharu Koike, Ludovico Minati.

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
