## [Decision Letter · Decision Letter 0]

9 Jul 2020

PONE-D-20-13446

Behavioral and physiological correlates of kinetically tracking a chaotic target

PLOS ONE

Dear Dr. Takagi,

Thank you for submitting your manuscript to PLOS ONE. After careful consideration, we feel that it has merit but does not fully meet PLOS ONE’s publication criteria as it currently stands. Therefore, we invite you to submit a revised version of the manuscript that addresses the points raised during the review process.

ACADEMIC EDITOR:  The reviewer 1 has strong concerns on your study. Please provide responses to the reviewer 1 to justify the significance of your study.

We look forward to receiving your revised manuscript.

Kind regards,

Kei Masani

Academic Editor

PLOS ONE

Journal Requirements:

Reviewers' comments:

Reviewer's Responses to Questions

**Comments to the Author**

1. Is the manuscript technically sound, and do the data support the conclusions?

Reviewer #1: Partly

Reviewer #2: Yes

2. Has the statistical analysis been performed appropriately and rigorously? 

Reviewer #1: Yes

Reviewer #2: Yes

3. Have the authors made all data underlying the findings in their manuscript fully available?

Reviewer #1: Yes

Reviewer #2: No

4. Is the manuscript presented in an intelligible fashion and written in standard English?

Reviewer #1: Yes

Reviewer #2: Yes

5. Review Comments to the Author

Reviewer #1: The authors investigated the tracking performance towards a time-varying target with chaotic dynamics (if I follow the authors’ indication). They simultaneously observed diverse measures, behavioral and neurophysiological ones.

Honestly, I cannot still find any significant, interesting, and novel points in this study, at least in the current style. Please respond carefully to the following comments.

Major:

Although the authors suggested “a chaotic target,” is it truly chaotic in two-dimensional space? To my knowledge, chaotic dynamics should have nonlinearity and more than three dimensions in a continuous-time system (NOTE: two-dimensional and nonlinear system can generate chaos in a discrete-time system). Of course, you will answer to this question as “Rossler equation can generate chaos,” but I am totally skeptical about the chaotic properties. It is not self-evident that nonlinear and chaotic three-dimensional trajectories still maintain its chaotic property after being projected into a two-dimensional plane. You should carefully check whether the two-dimensional target trajectories still retain chaotic properties while calculating some measured to validate chaotic property.

Along with the previous major comment, it is totally doubtful whether the authors have investigated the influence of the “level of the chaoticity” on the motor and neurophysiological responses.

Besides the chaotic property, the authors should calculate the autocorrelation of the target trajectories. The autocorrelation can reflect the tendency of tracking error because the larger autocorrelation yields higher predictability of the target position.

I cannot understand why the authors reported diverse behavioral measures, such as position, velocity, acceleration, and force using different measures. In performing a statistical comparison for multiple times, you can easily find at least one significant difference with a high possibility. Additionally, it is possible to calculate the target position, velocity, and acceleration. Why not calculate the target error in position, velocity, and acceleration? These can be a more direct measure for tracking performance. For force, it may be more interesting to discuss it simultaneously with EMG while simultaneously showing the learning curve of force and co-contraction (of course, several studies have already reported such perspective).

I cannot understand why the authors reported diverse neurophysiological measures, such as EEG, EMG, and respiratory response. In performing a statistical comparison for multiple times, you can easily find at least one significant difference with a high possibility. No confirmation of the relationship among different measures. I cannot find any rational reason to measure and report diverse measures.

Reviewer #2: The manuscript provides a valuable extension to the body of knowledge surrounding chaotic dynamics and human control, particularly in its breadth of data collected. It is on this basis, and its excellent presentation that I recommend its publication.

I did not find any areas that required revision, although there was at least one point on which I had questions. In Figure 2A, the human trajectory for the high chaoticity condition shows intriguing differences from the Rossler dynamic. It appears as though there are two folds, or that the dynamics were nonstationary such that the fold location was moving during the trial (which seems very likely). I would be very interested to see some analysis on the human trajectories themselves to understand how well the embedding dimension, number of fixed points, or other standard measures matched between target and human. The study is already wide ranging in its analyses, so I will leave it up to the editor and the authors' own curiosity as to whether such an analysis should be included in this particular manuscript.

6. PLOS authors have the option to publish the peer review history of their article (what does this mean?). If published, this will include your full peer review and any attached files.

Reviewer #1: No

Reviewer #2: No

---

## [Author Response · Author response to Decision Letter 0]

10 Aug 2020

We thank the editor and reviewers for the care with which they have examined our paper, and for their constructive suggestions. We describe below how we have addressed each of the comments. The changes are highlighted in blue in the revised manuscript.

Reviewer #1

The authors investigated the tracking performance towards a time-varying target with chaotic dynamics (if I follow the authors’ indication). They simultaneously observed diverse measures, behavioral and neurophysiological ones.

Honestly, I cannot still find any significant, interesting, and novel points in this study, at least in the current style. Please respond carefully to the following comments.

Major:

Although the authors suggested “a chaotic target,” is it truly chaotic in two-dimensional space? To my knowledge, chaotic dynamics should have nonlinearity and more than three dimensions in a continuous-time system (NOTE: two-dimensional and nonlinear system can generate chaos in a discrete-time system). Of course, you will answer to this question as “Rossler equation can generate chaos,” but I am totally skeptical about the chaotic properties. It is not self-evident that nonlinear and chaotic three-dimensional trajectories still maintain its chaotic property after being projected into a two-dimensional plane. You should carefully check whether the two-dimensional target trajectories still retain chaotic properties while calculating some measured to validate chaotic property.

We first would like to thank the Reviewer for their frank and helpful comments.

We feel the need to respectfully point out that one fundamental aspect of chaotic dynamics is that, regardless of the structural dimensionality of the underlying system (number of variables), it is generally visible in all state variables. This a well-known and established aspect that underlines the majority of techniques which are used to analyse non-linear time-series and, indeed, detect chaos. The topic is extensively discussed, for example, in the following reference, which has been added to the paper "H. Kantz and T. Schreiber. Nonlinear Time Series Analysis. Cambridge University Press; 1997". An important realization of the concept is Taken's theorem, a fundamental theorem in non-linear dynamics which implies that chaotic attractors can generally be reconstructed from time-lagged measurements of a single variable, in other words replacing variables such as (x,y,z) with x(t), x(t-d), x(t-2d) etc. and obviously implying that chaos is also evident when two state variables are considered; the following reference has been added to the paper "F. Takens, ‘‘Detecting strange attractors in turbulence,’’ in Lecture Notes in Mathematics., vol. 898. New York, NY, USA: Springer, 1981, pp. 366–381."

We also feel the need to respectfully point out that it is a well-established fact that chaotic dynamics should have nonlinearity and more than two (not three) dimensions in a continuous-time system: indeed, the most well-known chaotic systems such as the Roessler system, the Lorenz system and so on have three, not more variables. This is abundantly discussed in references such as "Ott E. Chaos in Dynamical Systems. Cambridge University Press; 2002." and "R. C. Hilborn. Chaos and Nonlinear Dynamics: An Introduction for Scientists and Engineers. Oxford University Press; 1994.", which are now cited in the paper.

Along with the previous major comment, it is totally doubtful whether the authors have investigated the influence of the “level of the chaoticity” on the motor and neurophysiological responses.

We completely agree with the reviewer that for a study of this kind, it is important to carefully check and explicitly show that the trajectories are indeed chaotic.

Along with the previous major comment, it is totally doubtful whether the authors have investigated the influence of the “level of the chaoticity” on the motor and neurophysiological responses. For this purpose, we have added analyses of the largely Lyapunov exponent and correlation dimension for the x and y coordinate time-series. The reviewer is invited to refer to the newly added Table 1, wherein we report the largest Lyapunov exponent and the correlation dimension for each setting of the parameter a: both parameters unquestionably demonstrate that, elevating a, the dynamics become increasingly chaotic. The following sentences have been added to the manuscript under the "Task design" subsection "On this basis, the largest Lyapunov exponent λ_MAX and correlation dimension D_2 can be readily calculated even from the separate x and y time-series. As documented in Table 1, for a=0.05, one has λ_MAX<0 and D_2≈1, indicating period dynamics; for a>0.15, both measures monotonically increase until λ_MAX≈0.07 and D_2≈2, hallmarking the low-dimensional chaotic dynamics that knowingly characterize this attractor [31-33]."

Besides the chaotic property, the authors should calculate the autocorrelation of the target trajectories. The autocorrelation can reflect the tendency of tracking error because the larger autocorrelation yields higher predictability of the target position.

We agree with the reviewer on this point also, it is indeed important to show the autocorrelation function. We have calculated them separately for the x and y coordinates and all 4 levels of the parameter a, and they are visible in Fig 3. The following text has been added to the "Task design" subsection "Accordingly, the autocorrelation functions, which initially oscillate between ±1, decay faster with increasing a, representing the loss of periodicity (Fig 3)”.

I cannot understand why the authors reported diverse behavioral measures, such as position, velocity, acceleration, and force using different measures. In performing a statistical comparison for multiple times, you can easily find at least one significant difference with a high possibility. Additionally, it is possible to calculate the target position, velocity, and acceleration. Why not calculate the target error in position, velocity, and acceleration? These can be a more direct measure for tracking performance. For force, it may be more interesting to discuss it simultaneously with EMG while simultaneously showing the learning curve of force and co-contraction (of course, several studies have already reported such perspective).

Thank you for this comment. Indeed, the probability of making a significant finding increases with the number of tests. This is controlled by Tukey’s HSD, which controls for the family-wise error rate. All multiple comparisons were made using Tukey’s HSD. In line 106, we have added “Tukey’s HSD” to refer to the multiple comparisons test employed to account for multiple post-hoc tests. Correcting for multiple tests across different variables is typically not done, as they are effectively independent.

The measures we selected for the analysis were not chosen haphazardly. A statistical analysis was conducted on the tangential velocity for two reasons. First, to show how the participant’s behaviour changed as a function of the control parameter setting (to examine how the task’s difficulty was reflected in the speed of the movement). Second, an increase in the tangential velocity over time is a plausible signature of motor learning as submovements (intermittent movement with stops in between) decrease with practice [1]. 

The rationale for the analysis on the target acceleration magnitude was to highlight the difference in the target’s acceleration under different control parameter settings due to the folding, which caused abrupt increases in the acceleration magnitude (Fig 3A). This was used to illustrate the effect of the control parameter setting on the target’s trajectory i.e., the increased frequency (occurrence) and size of the folding, and to illustrate how the participant exerted a force in reaction to the folding, following on to the measure of movement delay.

We conducted precisely the analysis on the force that the reviewer suggests. Fig 4A shows the normalized force magnitude as a function of the block number, and Fig 5A shows the normalized total EMG as a function of the block number. While the EMG decreased over time for all control parameter settings, the force magnitude actually increased for some control parameter settings. Thus, the reduction in EMG was likely due to a decrease in co-contraction, and not to a reduction in the exerted force.

I cannot understand why the authors reported diverse neurophysiological measures, such as EEG, EMG, and respiratory response. In performing a statistical comparison for multiple times, you can easily find at least one significant difference with a high possibility. No confirmation of the relationship among different measures. I cannot find any rational reason to measure and report diverse measures.

The EEG, EMG and respiration in themselves are worthy of study, and their interpretation requires their conjoint measurement. 

Studies have reported how the power in the delta and theta bands increase at the onset of motor imagery [2], to the processing of errors [3] and for sustained attention [4]. The EEG measurements suggest that a larger control parameter setting evoked increased involvement in the thalamus, cortex and the cerebellum, which plays an important role in predicting future states of a target trajectory [5,6]. This is a finding made possible only through the measurement of EEG.

The EMG measurements revealed the change in the motor behaviour of our participants as they learned to track the target driven by chaotic motion. The EMG decreased with the block number for all control parameter settings, even though the exerted force actually increased for some settings. This suggests a decrease in muscular co-contraction, which is related to the learning of a model of the novel chaotic dynamics [7,8]. This interpretation is supported by the EEG findings (namely the increased involvement of the cerebellum, which is critical to predicting future states).

The synchronization between respiration and voluntary motion is well-established [9–12], but the degree to which the synchronization depends on the difficulty of the motor task was unclear. We found that the synchronization between the respiration and the movement increased with the control parameter setting. This synchronization could have been due to an increase in sustained attention and focus, as revealed by the EEG measurements.

Thus, the EEG, EMG and respiration complement one another in interpreting the change in our participants’ behaviour when tracking a target governed by chaotic motion. Furthermore, each measurement offers unique insight into the physiological and behavioural changes that occur when attempting to predict increasingly chaotic dynamics.

Reviewer #2

The manuscript provides a valuable extension to the body of knowledge surrounding chaotic dynamics and human control, particularly in its breadth of data collected. It is on this basis, and its excellent presentation that I recommend its publication.

I did not find any areas that required revision, although there was at least one point on which I had questions. In Figure 2A, the human trajectory for the high chaoticity condition shows intriguing differences from the Rossler dynamic. It appears as though there are two folds, or that the dynamics were nonstationary such that the fold location was moving during the trial (which seems very likely). I would be very interested to see some analysis on the human trajectories themselves to understand how well the embedding dimension, number of fixed points, or other standard measures matched between target and human. The study is already wide ranging in its analyses, so I will leave it up to the editor and the authors' own curiosity as to whether such an analysis should be included in this particular manuscript.

Thank you for the comment. The variability in the participant’s cursor position gives the appearance of multiple folds, but only a single peak in the force magnitude was observed, corresponding to one movement in reaction to the fold.

We were also curious as to whether our participants followed the location of the fold, which varied over time, rather than simply following the average motion of the fold over an entire trial. This was the motivation behind the surrogate error (eq. 4), where the cursor’s amplitude was averaged over an entire trial. If the surrogate error and the tracking error were comparable with a high control parameter setting, then the participants likely did not follow the fold location per se, but only tracked the phase of the limit cycle orbit. However, we found that the surrogate error was significantly greater than the tracking error, suggesting that participants may have been tracking the fold location that varied over time too.

References

1. Pratt J, Abrams RA. Practice and Component Submovements: The Roles of Programming and Feedback in Rapid Aimed Limb Movements. J Mot Behav. 1996;28: 149–156. doi:10.1080/00222895.1996.9941741

2. Barios JA, Ezquerro S, Bertomeu-Motos A, Nann M, Badesa FJ, Fernandez E, et al. Synchronization of Slow Cortical Rhythms During Motor Imagery-Based Brain-Machine Interface Control. Int J Neural Syst. 2019;29: 1850045. doi:10.1142/S0129065718500454

3. Cohen MX, van Gaal S. Subthreshold muscle twitches dissociate oscillatory neural signatures of conflicts from errors. NeuroImage. 2014;86: 503–513. doi:10.1016/j.neuroimage.2013.10.033

4. Clayton MS, Yeung N, Cohen Kadosh R. The roles of cortical oscillations in sustained attention. Trends Cogn Sci. 2015;19: 188–195. doi:10.1016/j.tics.2015.02.004

5. Wolpert DM, Miall RC, Kawato M. Internal models in the cerebellum. Trends Cogn Sci. 1998;2: 338–347. doi:10.1016/S1364-6613(98)01221-2

6. Ito M. Control of mental activities by internal models in the cerebellum. Nat Rev Neurosci. 2008;9: 304–313. doi:10.1038/nrn2332

7. Thoroughman KA, Shadmehr R. Electromyographic Correlates of Learning an Internal Model of Reaching Movements. J Neurosci. 1999;19: 8573–8588. doi:10.1523/JNEUROSCI.19-19-08573.1999

8. Franklin DW, Burdet E, Tee KP, Osu R, Chew C-M, Milner TE, et al. CNS Learns Stable, Accurate, and Efficient Movements Using a Simple Algorithm. J Neurosci. 2008;28: 11165–11173. doi:10.1523/JNEUROSCI.3099-08.2008

9. Daffertshofer A, Huys R, Beek PJ. Dynamical coupling between locomotion and respiration. Biol Cybern. 2004;90: 157–164. doi:10.1007/s00422-004-0462-x

10. Hoffmann CP, Bardy BG. Dynamics of the locomotor-respiratory coupling at different frequencies. Exp Brain Res. 2015;233: 1551–1561. doi:10.1007/s00221-015-4229-5

11. Sporer BC, Foster GE, Sheel AW, McKenzie DC. Entrainment of breathing in cyclists and non-cyclists during arm and leg exercise. Respir Physiol Neurobiol. 2007;155: 64–70. doi:10.1016/j.resp.2006.02.013

12. Schmid M, Conforto S, Bibbo D, D’Alessio T. Respiration and postural sway: detection of phase synchronizations and interactions. Hum Mov Sci. 2004;23: 105–119. doi:10.1016/j.humov.2004.06.001

---

## [Decision Letter · Decision Letter 1]

8 Sep 2020

Behavioral and physiological correlates of kinetically tracking a chaotic target

PONE-D-20-13446R1

Dear Dr. Takagi,

We’re pleased to inform you that your manuscript has been judged scientifically suitable for publication and will be formally accepted for publication once it meets all outstanding technical requirements.

Kind regards,

Kei Masani

Academic Editor

PLOS ONE

Additional Editor Comments (optional):

Reviewers' comments:

Reviewer's Responses to Questions

**Comments to the Author**

1. If the authors have adequately addressed your comments raised in a previous round of review and you feel that this manuscript is now acceptable for publication, you may indicate that here to bypass the “Comments to the Author” section, enter your conflict of interest statement in the “Confidential to Editor” section, and submit your "Accept" recommendation.

Reviewer #1: All comments have been addressed

Reviewer #2: All comments have been addressed

2. Is the manuscript technically sound, and do the data support the conclusions?

Reviewer #1: Partly

Reviewer #2: Yes

3. Has the statistical analysis been performed appropriately and rigorously? 

Reviewer #1: Yes

Reviewer #2: Yes

4. Have the authors made all data underlying the findings in their manuscript fully available?

Reviewer #1: Yes

Reviewer #2: No

5. Is the manuscript presented in an intelligible fashion and written in standard English?

Reviewer #1: Yes

Reviewer #2: Yes

6. Review Comments to the Author

Reviewer #1: The authors responded to the reviewers' comments to some degree. For acceptance, I have no doubt.

Based on the Lyapunov exponent, I found significant chaotic dynamics only in two conditions. I cannot be sure whether the authors examined the influence of the level of chaotic dynamics on motor performance and related physiological features based on only two kinds of chaotic dynamics. It sounds to be exaggerated. Except for this point, there is no problem with the decision as to the acceptance.

Reviewer #2: After my last very brief review, I had hoped to give a more in depth review here. Again, however, I find that the analyses are presented soundly and clearly, which, given the charter of PLOS ONE, translates to few comments on my part.

Again, the authors present the results of a motor tracking study using a chaotically varying target. Concomitant with the focus on chaotic systems, a wide range of behavioral metrics were analyzed for their relation to the chaotic target and the subject's tracking performance. Indeed, as is tradition for chaotic systems, the chaoticity tends to show up in all parts of a coupled system. The appropriate way to interpret such results is as an extension of the body of knowledge of such systems.

I disagree, if I read the discussion correctly, that no known method or algorithm exists that could model the participant's state and estimation error. For instance, I direct attention towards Voss, H. U. (2018). A delayed-feedback filter with negative group delay. Chaos: An Interdisciplinary Journal of Nonlinear Science, 28(11), 113113. If I have understood the authors correctly, then they may wish to reference such methods.

Finally, less of a comment and more of a hope for future studies. I was excited to see the use of EMG recordings, but a bit disappointed that more analysis was not aimed at the temporal unfolding and likely anticipation of tracking movements. If I may exploit this arena for doing so, I suggest looking more deeply there, and recording from muscle groups in the back and legs, which are likely to show preparatory activation well before the 170 ms standard.

7. PLOS authors have the option to publish the peer review history of their article (what does this mean?). If published, this will include your full peer review and any attached files.

Reviewer #1: No

Reviewer #2: No

---

## [Editor Report · Acceptance letter]

10 Sep 2020

PONE-D-20-13446R1 

Behavioral and physiological correlates of kinetically tracking a chaotic target 

Dear Dr. Takagi:

I'm pleased to inform you that your manuscript has been deemed suitable for publication in PLOS ONE. Congratulations! Your manuscript is now with our production department. 

Kind regards, 

on behalf of

Dr. Kei Masani 

Academic Editor

PLOS ONE